# Dissecting the Long-Term Effect of Stress Early in Life on *FKBP5*: The Role of miR-20b-5p and miR-29c-3p

**DOI:** 10.3390/biom14030371

**Published:** 2024-03-19

**Authors:** Nadia Cattane, Maria Grazia Di Benedetto, Ilari D’Aprile, Marco Andrea Riva, Annamaria Cattaneo

**Affiliations:** 1Biological Psychiatry Unit, IRCCS Istituto Centro San Giovanni di Dio Fatebenefratelli, 25125 Brescia, Italy; ncattane@fatebenefratelli.eu (N.C.); maria.dibenedetto@unimi.it (M.G.D.B.); ilari.daprile@unimi.it (I.D.); m.riva@unimi.it (M.A.R.); 2Department of Pharmacological and Biomolecular Sciences, University of Milan, 20133 Milan, Italy

**Keywords:** early-life stress, cortisol, glucocorticoid receptors, *FKBP5*, miR-20b-5p, miR-29c-3p

## Abstract

Exposure to early-life stress (ELS) has been related to an increased susceptibility to psychiatric disorders later in life. Although the molecular mechanisms underlying this association are still under investigation, glucocorticoid signaling has been proposed to be a key mediator. Here, we used two preclinical models, the prenatal stress (PNS) animal model and an in vitro model of hippocampal progenitor cells, to assess the long-term effect of ELS on *FKBP5*, *NR3C1*, *NR3C2*, and *FoxO1*, four stress-responsive genes involved in the effects of glucocorticoids. In the hippocampus of male PNS rats sacrificed at different time points during neurodevelopment (PND 21, 40, 62), we found a statistically significant up-regulation of *FKBP5* at PND 40 and PND 62 and a significant increase in *FoxO1* at PND 62. Interestingly, all four genes were significantly up-regulated in differentiated cells treated with cortisol during cell proliferation. As *FKBP5* was consistently modulated by PNS at adolescence (PND 40) and adulthood (PND 62) and by cortisol treatment after cell differentiation, we measured a panel of miRNAs targeting *FKBP5* in the same samples where *FKBP5* expression levels were available. Interestingly, both miR-20b-5p and miR-29c-3p were significantly reduced in PNS-exposed animals (both at PND40 and 62) and also in the in vitro model after cortisol exposure. Our results highlight the key role of miR-20b-5p and miR-29c-3p in sustaining the long-term effects of ELS on the stress response system, representing a mechanistic link possibly contributing to the enhanced stress-related vulnerability to mental disorders.

## 1. Introduction

Exposure to early-life stress (ELS) has been associated with an increased risk for both physical and mental illnesses at adulthood [1]. While the underlying mechanisms remain poorly understood, glucocorticoid signaling, whose alterations have been largely associated with psychopathology [2], has been suggested to be among the key biological systems negatively affected by ELS [3,4,5].

Glucocorticoid hormones (GCs; cortisol in humans and corticosterone in rodents) are powerful steroids released in response to stress. Although they are essential for brain maturation, clinical observations and rodent studies have widely demonstrated that elevated GC levels, following ELS exposure, can cause long-lasting alterations in cognition and cortical thickness, altering the functioning of stress-related neurobiological systems and contributing to the development of several mental disorders later in life [6,7].

GCs are released in the systemic blood flow and reach multiple organs in the body, including the brain. They exert their effects through two types of receptors, the glucocorticoid receptor (GR), encoded by the *NR3C1* gene, and the mineralocorticoid receptor, encoded by the *NR3C2* gene. Both are cytoplasmic receptors that, in physiological conditions, translocate into the nucleus when activated by GCs and bind to specific DNA sequences, called glucocorticoid response elements (GREs), which are enhancers or repressors of gene transcription [4]. However, following ELS exposure, activated GRs, due to elevated levels of GCs, increase the transcription of the co-chaperonine FKBP Prolyl Isomerase 5 (*FKBP5*), which represses GR activity by limiting the translocation of the receptor complex into the nucleus and by inhibiting the transcription of its target genes [8,9]. Specifically, preclinical studies suggest that an increased expression of *FKBP5* in several brain regions is associated with decreased stress coping, increased stress responsiveness, increased anxiety, and impaired fear extinction, all traits associated with vulnerability to psychiatric disorders (see for review [4,10]). In line with preclinical findings, post-mortem human brain studies also report elevated *FKBP5* mRNA levels in cortical regions across different psychiatric disorders, including depression [11], post-traumatic stress disorder (PTSD) [12], and schizophrenia [13].

Another interesting gene that is associated with *NR3C1* and is also modulated by ELS is Forkhead Box O1 (*FoxO1*) (see Figure 1). Our group also showed that it is up-regulated in the total hippocampus of adult rats exposed to prenatal stress (PNS) as well as in peripheral blood samples of control subjects characterized by childhood trauma [14]. Interestingly, upon testing for gene x environment (GxE) interactions in two Genome wide association studies (GWAS) samples of adult controls and depressed patients exposed to childhood traumatic experiences, *FoxO1* has been confirmed to be a stress responsive gene, able to mediate the effect of the environment on the development of depressive symptoms [14]. Accordingly, *FoxO1* has been found to be able to mediate the effect of GCs in other disorders, including cancer [15] and insulin resistance [16,17].

Although ELS exposure can contribute to the onset of psychopathology or to the development of clinical symptoms later in life, through the involvement of stress-responsive genes, the underlying and long-lasting GR-dependent mechanisms induced by ELS are still unclear. Therefore, the characterization of these alterations is extremely important to increase the knowledge and to develop novel targets in the field of stress-related psychiatric disorders.

In this regard, during recent years, both in vivo and in vitro models have attracted a great deal of interest because of their ability to model the effects of stress in controlled conditions. Indeed, they allow for the investigation of stress response mechanisms by overcoming many of the ethical and technical/confounding issues related to human post-mortem brains, including the cause of death, agonal state, post-mortem delay, temperature of storage, procedures of tissue preservation, and chronic drug treatment [18,19]. Indeed, animal models are useful preclinical tools for studying the pathogenesis of mental disorders and the effectiveness of their treatment. While it is not possible to mimic all symptoms occurring in humans, it is however possible to investigate the behavioral, physiological, and neuroanatomical alterations relevant for these complex disorders in controlled conditions and in genetically homogeneous populations [20]. As an example, PNS in rodents is a well-characterized paradigm of stress early in life, able to induce long-lasting consequences in the offspring, including alterations at molecular and behavioral levels and in cognitive functions [21,22]. Specifically, PNS has been suggested to have long-lasting effects on the adult offspring’s hypothalamic–pituitary–adrenal (HPA) axis, with a persistent hyperactivation of the system [23]. However, different results have been reported also depending on the type of stress and on the timing of stress exposure.

Other than animal models, in the last decades, neuronal cell cultures have been extensively used to investigate the neurobiological effects of GCs, with most of the in vitro studies focused on the downstream effects of GR activation [24]. The main advantage of these cell cultures is the possibility to mimic human brain physiology while observing individual cell systems in isolation and in a manner that is highly reproducible as well as cost effective [25]. Moreover, they can be useful to demonstrate the causal relationship between a certain stressor and specific biological alterations which could be pivotal for the development of a specific mental disorder and therefore for testing novel drugs or preventive compounds.

Against this background, in the current paper, we used two different preclinical models to investigate the short- and long-lasting effects of stress on the expression levels of GR-related genes and to investigate the role of their targeting microRNAs (miRNAs) in such long-lasting effects. In particular, we measured *FKBP5*, *NR3C1*, *NR3C2,* and *FoxO1* genes in the hippocampus of male rats exposed or not to PNS and sacrificed at different neurodevelopmental time points (PNDs). We then validated the results in an in vitro model represented by human hippocampal progenitor cells that we treated with the stress hormone cortisol during 3 days of proliferation, and then we let cells differentiate into mature neurons for 21 days. Finally, we focused the attention on miRNAs as possible epigenetic mechanisms underlying the long-term effect of stress on the stress response system.

## 2. Materials and Methods

### 2.1. Prenatal Stress Model

Adult female and male Sprague Dawley rats were pair-housed with a same-sex conspecific with food and water available ad libitum (21 ± 1 °C, 60 ± 10% relative humidity, reversed 12/12 h light/dark cycle). After 10 days, rats were mated for 24 h and individually housed immediately thereafter. Pregnant females were randomly assigned to delivery control (CTRL; n = 8) and prenatal stress (PNS; n = 8) conditions. PNS procedure was performed daily in three stress sessions at different time points spaced out by 2–3 h, in order to reduce habituation to repeated restraint stress. Briefly, the stress session consisted of restraining pregnant dams in a transparent Plexiglas cylinder (7.5 cm diameter, 19 cm length) under bright light (6500 lux) for 45 min during the last week of gestation, such as from gestational day (GD) 14 to delivery [26,27]. Control pregnant females were left undisturbed in their home cages. On postnatal day (PND) 1, pups from CTRL and PNS dams were weighted, and litters were culled to 5 males and 5 females. Weaning occurred on PND 21, and same-sex rats were housed in groups of 3 per cage. For this experiment, which focused on adult animals exposed or not to PNS, 1 or maximum 2 animals per litter were considered for the analyses. Control rats were left undisturbed. Male offspring from control and PNS groups were killed at postnatal day (PND) 21, 40, and 62 for the whole dissection of brain areas including the hippocampus [27]. Rat handling and experimental procedures were performed in accordance with the EC guidelines (EC Council Directive 86/609 1987) and with the Italian legislation on animal experimentation (D.L. 116/92), in accordance with the National Institute of Health Guide for the Care and Use of Laboratory Animals. The experiments on animal models were approved by the ethical committee—no humans studies are included. Ethical committee name: Italian Ministry of Health—Direzione generale della Sanità animale e dei farmaci veterinari. Approval code: 752/2020-PR; approval date: 28 July 2020.

### 2.2. In Vitro Cell Model

The immortalized, multipotent, human hippocampal progenitor cell line HIP-009 (propriety of PhoenixSongs Biologicals, Branford, CT, USA) was used for all the experiments. Cells were expanded and differentiated as described in the manufacturer’s instructions. Briefly, HIP-009 cells were proliferated on 10 μg/mL laminin-coated (Merck-Millipore, Burlington, MA, USA) dishes in Dulbecco’s modified Eagle’s medium (DMEM). Subsequently, cells were seeded and grown on these dishes in Neural StemCell Growth Medium (PhoenixSongs Biologicals), with added growth medium supplement cocktail (Neural StemCell Growth Supplement, 10 ng/mL basic fibroblast growth factor (bFGF), 20 ng/mL epidermal growth factor (EGF), 3 μM CHIR-99021, and 1 μg/mL laminin) [28]. Proliferating cells were split for expansion every 4 or 5 days. Before the start of differentiation, expanded cells were grown in Neural Transition Medium (PhoenixSongs Biologicals) with added transition medium supplement cocktail (Neural Transition Supplement, 10 ng/mL bFGF, 20 ng/mL EGF, and 1 μg/mL laminin) and cultured for 3 days. After that, cells were differentiated in Neural Differentiation Medium (PhoenixSongs Biologicals) with added differentiation medium supplement cocktail (Neural Differentiation Supplement, 20 ng/mL brain-derived neurotropic factor (BDNF), 20 ng/mL glial cell-derived neurotropic factor, and 1 μg/mL laminin). The differentiation process was performed for 21 days, during which half of the medium was changed every 3 days. All cell culture was performed at 37 °C and 5% CO_2_ in a humidified atmosphere.

We performed two different treatments:Short-term treatment: HIP-009 cells were treated for 3 days during proliferation with cortisol (100 μM) or the vehicle, and cells were harvested immediately after the end of the treatment.Long-term treatment: HIP-009 cells were treated with cortisol (100 μM) or the vehicle during 3 days of proliferation, grown in transition medium for 3 days, and then harvested after 21 days of differentiation without any treatment.

Five biological replicates were obtained for each treatment and condition (cortisol or vehicle).

By using these two in vitro paradigms, we aimed to mimic a direct exposure to ELS, in terms of cortisol treatment, during neurodevelopment (short-term treatment) and to evaluate whether such changes could be maintained over time, after differentiation into mature neurons (long-term treatment).

### 2.3. RNA Isolation

Total RNA was isolated from HIP-009 cells using the AllPrep DNA/RNA/miRNA kit (Qiagen, Hilden, Germany) from whole blood collected in PaxGene tubes using the PAXgene blood miRNA kit (Qiagen, Hilden, Germany) and from the rats’ brains using PureZol RNA isolation reagents (Bio-Rad Laboratories, Hercules, CA, USA), according to the manufacturer’s protocols. RNA quantity and quality were assessed by evaluation of the A260/280 and A260/230 ratios using a Nanodrop spectrophotometer (NanoDrop Technologies, Wilmington, DE, USA). All the RNA samples were analyzed for their integrity by using the Agilent Bioanalyzer (Agilent, Santa Clara, CA, USA), and all the RNA samples had an RNA Integrity Number (RIN) > 8.

### 2.4. Real-Time PCR

The expression levels of four candidate genes involved in stress response mechanisms, FKBP Prolyl Isomerase 5 (*FKBP5*), Nuclear Receptor Subfamily 3 Group C Member 1 (*NR3C1*), Nuclear Receptor Subfamily 3 Group C Member 2 (*NR3C2*), and Forkhead Box O1 (*FoxO1*), and the housekeeping gene Glyceraldehyde-3-Phosphate Dehydrogenase (*GAPDH*) were evaluated in human hippocampal progenitor cells treated with vehicle or cortisol. The mRNA levels of the same stress-related genes and of the housekeeping gene beta actin (*b-actin*) were also analyzed in the brain of PNS and control rats sacrificed at different time points during neurodevelopment. All RT-PCR analyses were conducted by using TaqMan Assays (Thermofisher, Waltham, MA, USA) on the CFX384 instrument (Bio-Rad Laboratories, Hercules, CA, USA), following the manufacturer’s instructions. All samples were assayed in triplicate.

The expression of *FKBP5* targeting miRNAs was analyzed by using TaqMan MicroRNA Assays (Thermofisher, Waltham, MA, USA) on the CFX384 instrument (Bio-Rad Laboratories, Hercules, CA, USA) following the manufacturer’s instruction.

Briefly, a total amount of 50 ng of the extracted total RNA, including miRNAs, from each sample was firstly reverse-transcribed using the TaqMan MicroRNA RT Kit (Thermofisher, Waltham, MA, USA) and then pre-amplified using TaqMan PreAmp Master Mix (Thermofisher, Waltham, MA, USA). The PreAmp product obtained was diluted with 0.1X TE and then evaluated for the expression levels of the miRNAs of interest by Real-Time PCR using the CFX384 instrument (Bio-Rad Laboratories, Hercules, CA, USA), the TaqMan MicroRNA Assays (Thermofisher, Waltham, MA, USA), and the TaqMan Universal Master Mix, no AmpErase UNG (Thermofisher, Waltham, MA, USA), following the manufacturer’s instructions. All the reactions were performed in triplicate. The relative expression levels of the miRNAs of interest were normalized to the mean levels of U6 and U87 for rats and RNU44 and RNU24 for human cells. The Pfaffl Method [29] or the 2^−DDCt^ [30] were used to determine the relative expression values of genes or miRNAs of interest.

### 2.5. Statistical Analyses

Expression data of stress-responsive genes are graphically expressed as means ± SEM. The significant differences among the groups were assessed with the Student’s *t*-test or ANOVA, and the statistical significance was set at *p*-value *<* 0.05. Gene expression and miRNA expression data were analyzed with ANOVA applying Bonferroni correction. All graphs were created by using GraphPad Prism 8.0.2.

## 3. Results

### 3.1. Effects of Prenatal Stress on the Modulation of Stress-Responsive Genes during Neurodevelopment

Our first aim was to investigate how stress early in life could affect the modulation of stress-responsive genes during neurodevelopment and to assess whether changes could persist up to adulthood. Thus, we measured the expression levels of *FKBP5*, *NR3C1*, *NR3C2,* and *FoxO1* in the hippocampus of male rats exposed or not to PNS and sacrificed at different postnatal days (PND 21, 40, and 62). Interestingly, we found a significant up-regulation of *FKBP5* expression levels in male PNS rats compared to controls sacrificed at PND 40 and 62 (+32%, *p* < 0.01 PND 40; +25%, *p* < 0.05 PND 62) and a trend of increase at PND 21 (+14%, *p* = 0.053). *FoxO1* was found to be significantly up-regulated only at PND 62 (+22%, *p* < 0.05) in PNS-exposed animals. A trend of decrease in *NR3C2* mRNA levels and a trend of increase in *NR3C1* expression were observed at PND 40 and PND 62 in PNS rats (see Figure 2).

### 3.2. Short- and Long-Term Effects of Cortisol Treatment in an In Vitro Model of Human Hippocampal Progenitor Cells

In order to better investigate the short- and long-term effects of stress on the modulation of genes involved in the stress response mechanisms in an alternative preclinical model, we used HIP-009 cells, and we treated them with the stress hormone cortisol at a concentration of 100 µM [31] during 3 days of cell proliferation. Then, we assessed its effects at the end of the third day of cell proliferation and after twenty-one days of cell differentiation by evaluating the expression levels of the same stress-responsive genes (*FKBP5*, *NR3C1*, *NR3C2,* and *FoxO1*) measured in the animal model of PNS. Interestingly, we found a statistically significant up-regulation of the four genes in treated and proliferating cells compared to vehicle-treated cells (2^−DDCt^ = 2.81, +181%, *p*-value = 2.73 × 10^−7^ for *FKBP5*; 2^−DDCt^ = 1.35, +35%, *p*-value = 0.007 for *NR3C1*; 2^−DDCt^ = 1.28, +28%, *p*-value = 0.025 for *NR3C2*; 2^−DDCt^ = 1.25, +25%, *p*-value = 0.002 for *FoxO1*) (Figure 3A). Interestingly, after 21 days of differentiation, all the genes maintained their up-regulation (2^−DDCt^ = 1.87, +87%, *p*-value = 3.03 × 10^−6^ for *FKBP5*; 2^−DDCt^ = 1.25, +25%, *p*-value = 0.032 for *NR3C1*; 2^−DDCt^ = 1.44, + 44%, *p*-value = 0.012 for *NR3C2*; 2^−DDCt^ = 1.12, +12%, *p*-value = 0.005 for *FoxO1*) in mature neurons, which were previously sensitized by cortisol treatment during 3 days of cell proliferation (Figure 3B).

### 3.3. Role of miRNAs Targeting FKBP5 in the Long-Lasting Effect of ELS

As *FKBP5* was the gene that was consistently modulated by PNS at PND 40 and 62 as well as in our in vitro cell model both during proliferation and also differentiation, we aimed at evaluating whether miRNAs targeting *FKBP5* could be involved in sustaining this long-term effect. Through a targeting analysis performed by using miRDB (https://mirdb.org/cgi-bin/search.cgi) (accessed 11 August 2023), we identified a panel of 20 validated miRNAs targeting *FKBP5*, which were miR-702-3p, miR-320-5p, miR-19b-1-5p, miR-19b-2-5p, miR-29, miR-133b-5p, miR-3541, miR-20b, miR-200a-3p, miR-141-3p, miR-17-1-3p, miR-203b-5p, miR-495, miR-335, miR-129-5p, miR-148a-3p, miR-148b-3p, miR-139-3p, miR-152-3p, and miR-3559-5p. Based on these bioinformatics analyses, we then measured their expression levels in the same samples where gene expression data on *FKBP5*, *NR3C1*, *NR3C2,* and *FoxO1* were available. Interestingly, we found a significant reduction for miR-20b-5p and miR-29c-3p in PNS animals both at PND 40 and PND 62 (−12%, *p* < 0.05 PND 40, −16%, *p* < 0.05 PND 62 for miR-20b-5p; −13%, *p* < 0.05 PND 40, −15%, *p* < 0.05 PND 62 for miR-29c-3p). An up-regulation was observed for miR-3559-5p and for miR-141-3p at the same PNDs (+21%, *p* < 0.05 PND 40, +24%, *p* < 0.05 PND 62 for miR-3559-5p; +24%, *p* < 0.05 PND 40, +27%, *p* < 0.05 PND 62 for miR-141-3p) (Figure 4).

As miR-20b-5p and miR-29c-3p were the most significant miRNAs whose down-regulation could explain the long-term up-regulation of *FKBP5*, we decided to validate these miRNAs also in our in vitro model of cells treated with cortisol (100 µM) during proliferation and that we let differentiate into mature neurons (to see whether they could mediate the long-term up-regulation of *FKBP5*). Interestingly, after 21 days of cell differentiation, we found a significant down-regulation of miR-20b-5p and miR-29c-3p levels (mean 2^−DDCt^ = 0.58, −42%, *p*-value = 0.004 for miR-20b-5p; mean 2^−DDCt^ = 0.90, −10%, *p*-value = 0.017 for miR-29c-3p) in mature neurons treated with cortisol during proliferation compared to those treated with the vehicle (Figure 5).

## 4. Discussion

In the current study, we not only confirmed the key role of *FKBP5* in the stress response mechanisms, especially in association with the long-lasting effects of stressful experiences early in life, but we also demonstrated the involvement of miR-20b-5p and miR-29c-3p in mediating the negative effect of stress early in life on *FKBP5* up-regulation.

We first investigated the negative effects of ELS on the expression levels of four selected genes involved in the stress response mechanisms, namely *FKBP5*, *NR3C1*, *NR3C2,* and *FoxO1*, by taking advantage of a well-established animal model of PNS. We measured their expression levels in the hippocampus of male PNS rats sacrificed at different time points during neurodevelopment (PND 21, 40, and 62), compared to control animals. Interestingly, we observed increased expression levels of *FKBP5* in PNS-exposed animals at PNDs 40 and 62 and also the presence of increased levels of *FoxO1* at adulthood, at PND 62, with a trend of increase at adolescence, at PND 40, suggesting that PNS can have long-term effects on *FKBP5*- and *FoxO1*-related signaling, in line with previous published data [14,32]. Indeed, for example, after exposing male and female mice overexpressing *FKBP5* in the brain to maternal separation for 14 days after birth, Criado-Marrero and collaborators found increased depressive-like behaviors and lower basal corticosterone levels, reflecting the role of *FKBP5* in the HPA axis functions, in aged mice [32]. The data on *FoxO1* also confirmed our previous published data, generated in another cohort of adult PNS rats, showing *FoxO1* among the differentially expressed and up-regulated genes found in the hippocampus of male adult rats (at PND 62) exposed to PNS and compared to control animals [14].

To corroborate the negative effects of stress on the expression levels of the four selected stress-responsive genes, we here used an in vitro model represented by human hippocampal progenitor cells that we treated with cortisol during cell proliferation. As expected, we observed a significant increase in the expression levels of all stress-related genes and, in particular, on FKBP5 in proliferating cells treated with cortisol, and interestingly, similar effects were also observed after 21 days of differentiation, without any treatment, in mature neurons previously sensitized by cortisol during proliferation. These results suggest that an exposure to a high concentration of cortisol, a condition that we have previously associated with a reduction in the hippocampal neurogenesis levels [14,31], is able not only to affect the expression levels of stress-responsive genes in an immature neuronal cellular model but also to cause alterations that persist over time, after cell differentiation into mature neurons. The increased expression levels of *NR3C1* and *NR3C2* both in the short- and the long-term treatments demonstrate the high demand for GRs in cells exposed to stressful challenges, in terms of exposure to GCs, and these findings confirm the already available literature data suggesting that the activated status of GRs is able to increase the transcription of the *FKBP5* gene [8,9]. Therefore, both in the short- and long-term treatments performed in our study, we can hypothesize an increased transcription of *FKBP5*, due to the activation of the GRs, the *NR3C1* and the *NR3C2*, which are up-regulated as well. Thus, our results suggest that, after cortisol treatment, the stress response mechanisms are hyperactivated and that this state of hyperactivation remains after cell differentiation, resulting in an altered mechanistic regulation of the stress response system, which in turn could contribute mechanistically to enhancing the vulnerability to psychiatric disorders.

As *FKBP5* was consistently modulated by PNS at different neurodevelopmental time points and by cortisol in the in vitro model, in order to dissect the potential mechanisms underlying the long-lasting modulation of *FKBP5*, we focused the attention on miRNAs as the possible epigenetic mediators.

After a bioinformatics analysis to select validated miRNAs targeting *FKBP5*, we measured 20 miRNAs in the hippocampus of PNS rats (where *FKBP5* expression levels were available) and we found that, among them, miR-20b-5p and miR-29c-3p were significantly down-regulated at both PND 40 and 62 in PNS male adult animals compared to controls. Interestingly, when we measured these two miRNAs in the in vitro model, we observed a significant down-regulation of both miR-20b-5p and miR-29c-3p levels in differentiated cells that were previously exposed to cortisol during cell proliferation, showing an opposite modulation compared to *FKBP5* mRNA levels.

MiR-20b-5p and miR-29c-3p, two miRNAs regulating cell differentiation and cell death, have been previously associated with psychiatric disorders, including depression and bipolar disorder [33]. For instance, alterations in miR-29c-3p levels have been found in extracellular vesicles (EVs) extracted from the brains of Flinders Sensitive Line rats, a well-validated animal model exhibiting depressive-like behaviors and glial (astrocytic) dysfunction [33], and in the prefrontal cortex (Brodmann area 9, BA9) of bipolar disorder patients [34]. They have been also linked with neurodegenerative diseases, above all with Alzheimer’s disease [35,36] and neurodevelopmental disorders, including schizophrenia [37] and autism spectrum disorder [38]. Overall, these studies suggest that miR-20b-5p and miR-29c-3p may represent promising predictive and diagnostic biomarkers and provide novel insights into the mechanisms by which miRNAs can bind to key target genes contributing to the pathogenesis of brain disorders. Therefore, persistent elevated levels of *FKBP5* along with lower levels of miR-20b-5p and miR-29c-3p in association with stress exposure in two different preclinical models could represent a mechanistic cause able to explain the increased vulnerability to developing psychiatric disorders in ELS-exposed subjects.

Importantly, our data show that such negative effects of stress, in terms of *FKBP5* up-regulation and miR-20b-5p and miR-29c-3p down-regulation, are already manifested during adolescence/early adulthood (PND 40-62). In line with the literature data, also coming from our research group, this period can indeed represent an important and critical temporal window of vulnerability to stress-related psychiatric disorders [27]. Indeed, we know that ELS exposures are responsible for an aberrant development of brain structure and functions, in particular of those brain regions involved in cognition and stress reactivity, such as the hippocampus [39]. Specifically, peripuberty/adolescence represents a transition period characterized by significant changes at hormonal and behavioral levels, as well as by continuous neurodevelopmental processes, which can contribute to both the sexual and social/cognitive maturation of the individual [40]. Therefore, exposure to stress early in life, namely during sensitive periods of development, may cause “biologically embedded” alterations. This means that the exposure to stressful experiences early in life causes long-lasting biological changes, altering the development and functioning of stress-related neurobiological systems and therefore contributing to the development of physical and mental disorders later in life [7]. One of the possible mechanisms mediating the negative effects of ELS and the observed neurodevelopmental consequences later in life could be represented by an excessive exposure to GCs, above the physiological levels. Although GCs are essential for brain maturation, the developing brain is vulnerable to excessive GCs, with reported long-lasting alterations in cognition and cortical thickness [6]. A recent paper by Provencal and collaborators has indeed suggested that early exposure to GCs changes the set point of future transcriptional responses to stress by inducing lasting changes in DNA methylation [6]. Similarly, large epidemiologic studies have suggested that increased prenatal exposure to synthetic GCs increases the risk for later psychiatric diagnoses [41,42,43].

## 5. Conclusions and Limitations

In conclusion, our results from the data integration of two different preclinical models indicate that PNS causes long-lasting up-regulation in the expression levels of *FKBP5* via the modulation of specific miRNAs, whose alterations become manifested already during peripuberty/adolescence and then persist over time. Our results highlight the key role of miR-20b-5p and miR-29c-3p in sustaining the long-term effects of stress early in life on the stress response system and possibly contributing to the enhanced vulnerability to developing a wide range of stress-related illnesses, including mental disorders. However, as we have not manipulated miR-20b-5p and miR-29c-3p in vitro, we cannot demonstrate their causal relationship with FKBP5 modulation under stress condition. Moreover, we know that the genes under investigation were chosen ahead of time, and therefore this study cannot be considered non-biased.

## Figures and Tables

**Figure 1 biomolecules-14-00371-f001:**
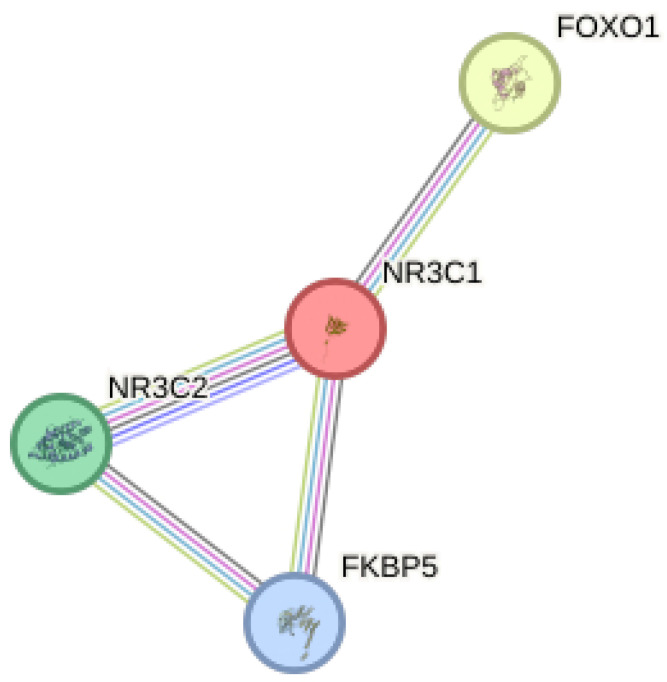
Protein–protein interaction network among the 4 stress-responsive genes (*FKBP5*, *NR3C1*, *NR3C2,* and *FoxO1*) obtained by using STRING (https://string-db.org/) (version 12.0, accessed on 18 November 2023).

**Figure 2 biomolecules-14-00371-f002:**
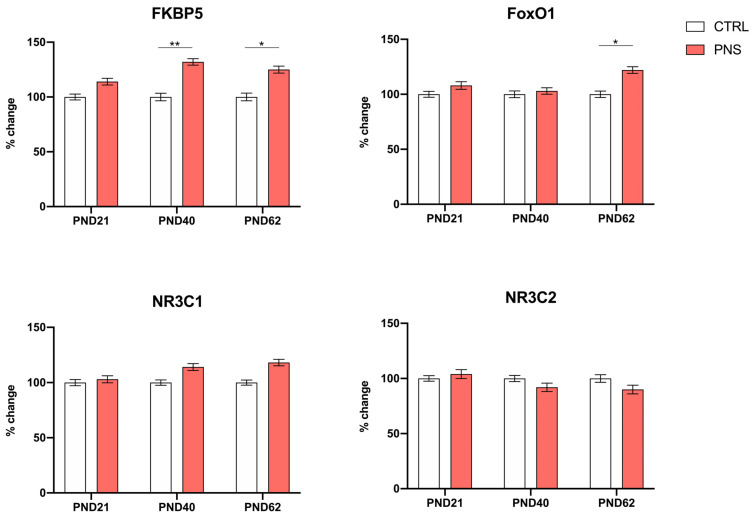
Expression levels of stress-responsive genes in the hippocampus of male PNS and control rats sacrificed at PND 21, 40, and 62. The data, expressed as the mean ± SEM, represent the % of change in *FKBP5*, *NR3C1*, *NR3C2,* and *FoxO1* normalized to beta actin levels and to the group of control rats (not exposed to PNS) at each respective age. * *p*-value < 0.05; ** *p*-value < 0.01.

**Figure 3 biomolecules-14-00371-f003:**
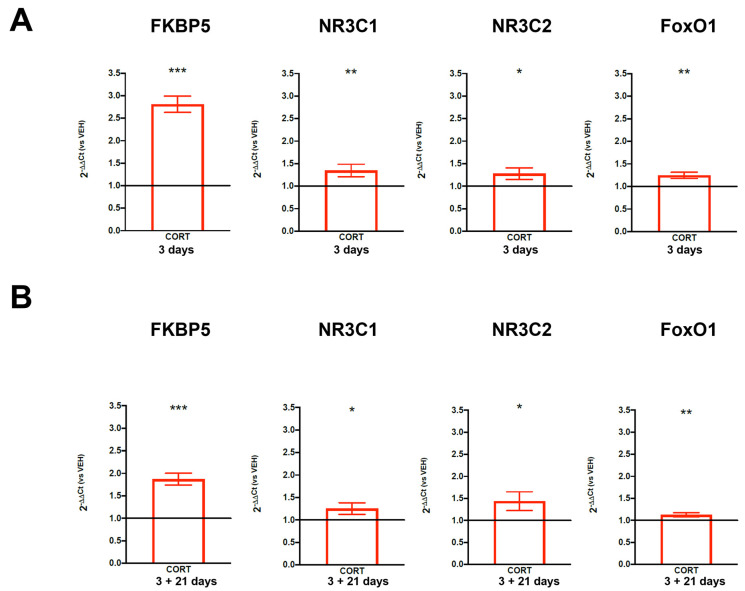
Expression levels of stress-responsive genes in human hippocampal progenitor stem cells. (**A**) shows the expression levels of *FKBP5*, *NR3C1*, *NR3C2,* and *FoxO1* in HIP-009 cells treated with cortisol or the vehicle during 3 days of cell proliferation. (**B**) shows the expression levels of *FKBP5*, *NR3C1*, *NR3C2,* and *FoxO1* in HIP-009 cells treated with cortisol or the vehicle during 3 days of proliferation and differentiated for 21 days without any treatment. The data, expressed as the mean ± SEM, represent 2^−DDCt^ and are normalized to GAPDH levels in cells receiving the vehicle, represented as the black line set at 1.0 on the Y axis. * *p*-value < 0.05; ** *p*-value < 0.01; *** *p*-value < 0.001.

**Figure 4 biomolecules-14-00371-f004:**
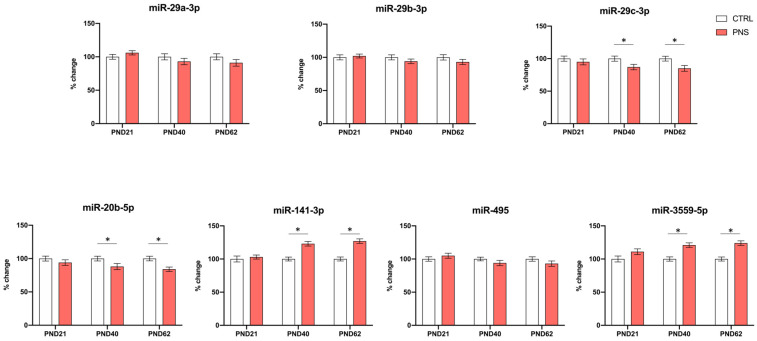
Expression levels of most interesting miRNAs targeting *FKBP5* (miR-29a-3p, miR-29b-3p, miR-29c-3p, miR-20b-5p, miR-141-3p, miR-495, miR-3559-5p) in the hippocampus of male PNS and control rats sacrificed at PND 21, 40, and 62. The data, expressed as the mean ± SEM, represent the % of change in miRNAs targeting *FKBP5* normalized to the mean levels of U6 and U87 and to the group of control rats (not exposed to PNS) at each respective age. * *p*-value < 0.05.

**Figure 5 biomolecules-14-00371-f005:**
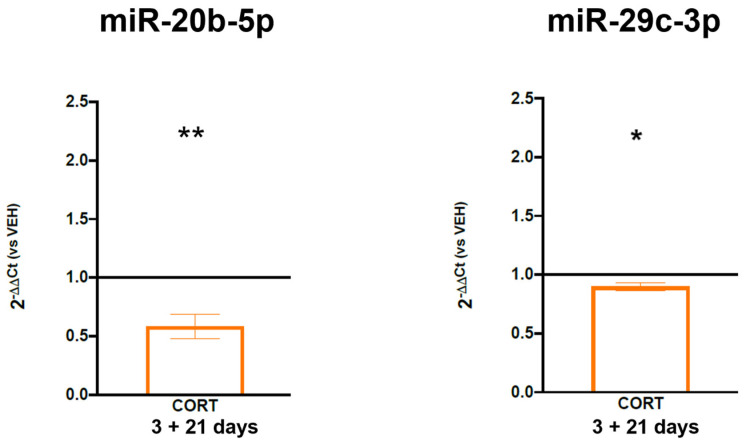
Expression levels of miR-20b-5p and miR-29c-3p in HIP-009 cells treated with cortisol or the vehicle during 3 days of proliferation and differentiated for 21 days without any treatment. The data, expressed as the mean ± SEM, represent 2^−DDCt^ and are normalized to the mean levels of RNU44 and RNU24 in cells receiving the vehicle, represented as the black line set at 1.0 on the Y axis. * *p*-value < 0.05; ** *p*-value < 0.01.

## Data Availability

The raw data supporting the conclusions of this article will be made available by the authors on request.

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
