# Peer review of "Dissecting the Long-Term Effect of Stress Early in Life on FKBP5: The Role of miR-20b-5p and miR-29c-3p"

_biomolecules, 2024, doi:10.3390/biom14030371_

Round 1

Reviewer 1 Report

Comments and Suggestions for Authors

In the manuscript by Cattane et al., the authors investigate in vivo and in vitro models of stress to examine the expression of several stress-responsive candidate genes (namely Fkbp5, Nr3c1, Nr3c2 and FoxO1) at selected times following the stress. The in vivo model consists of a restraint stress paradigm in which pregnant rat dams are restrained in a tube 3 times daily, although the authors omit information regarding which prenatal days the stress is started and ended. Typically, this model will stress dams from gestation days 7-8 through birth.  The focus of the study is on the offspring and the hippocampus is obtained on post-natal days (PND) 21, 40 and 62.

The in vitro model consists of hippocampal progenitor cells which are cultured and induced to differentiate along a neuronal path. For short term studies, these cells are treated with cortisol with a single administration of cortisol for 3 days during proliferation. The long term treatment, the cells are treated with cortisol for 3 days during proliferation, and then allowed to differentiate for an additional 21 days without further treatment.

In both models, RNA is isolated and analyzed using Taqman RT-PCR probes for the expression of the above noted stress responsive genes. The authors report the changes that occur in both models at the different time points.  They see changes in Fkbp5 and FoxO1 in vitro and in each of these mRNAs in the in vitro models. They next examine a series of miRNAs that target Fkbp5 to test whether changes in the levels of selected miRNAs might correlate with the increased levels of Fkbp5 mRNA. They show the expression levels of several of the miRNAs are reduced in vivo. Some of these are statistically significant, while others are reported as trends.  However, there is no statement in the text describing what qualifies a change to be trending and from the data it appears to be somewhat loose. In addition, they report changes in miR-20b-5p and miR-29c-3p in the long term treatment hippocampal culture paradigm.

Major Concern:

While the expression changes documented in the manuscript were done properly, it’s not clear that the observed changes in miRNAs are causal with respect to Fkbp5 expression.  To make this claim substantial, the authors would need to interfere with the expression of selected miRNAs (either in vitro or in vivo) and show a corresponding change in Fkbp5 mRNA. Since the authors preselected their candidate genes for the study, it is not surprising that they see changes in stress responsive genes. However, to conclude that the miRNAs are causal in this process, they would have to manipulate the levels of one or more miRNA and show an effect on Fkbp5.

Minor Concerns:

In materials and methods, the authors should state how many litters were represented in the prenatal restraint stressed and control groups. Because littermates tend to respond similarly to various conditions, it is important not to include littermates in the same experimental groups.

Regarding the data trends, the authors need to state in the statistic section how they mean trend and should report the associated significance (for example, miR-29a-30p and miR29b-3p and miR495 (in Fig. 4).  Some authors define findings that are trending at p<0.07 or so. Changes that simply go in the same direction are not necessarily trending, they may just not be significant.

In materials and methods, the authors are reminded to define which prenatal days the pregnant dams are stressed.

Is there a reason why different housekeeping genes are used in the two models?  Both G3pdh and Actb are often used for normalization, it is more accurate to use a housekeeping gene that shows expression levels comparable to the mRNA under study.

On p. 5, line 190, the term pre-amplificated is used instead of pre-amplified.

Reviewer 2 Report

Comments and Suggestions for Authors

I have read the manuscript Dissecting the Long-Term Effect of Stress Early in Life on 3 FKBP5: The Role of miR-20b-5p and miR-29c-3p., containing the results. The article can be published in your journal. However, the authors should take into account the following comments:
1. It is not clear whether the litters of animals were equalized in number; this is important for understanding the results;
2. The authors use beta-actin as a housekeeping gene. However, the constancy of the expression of this gene in ontogenesis is doubtful. The authors should study this issue and make corrections or changes;
3. The authors list ANOVA as one of the statistical methods used. However, it remains unclear what data are analyzed by ANOVA and what post-hoc test the authors use

Round 2

Reviewer 1 Report

Comments and Suggestions for Authors

The authors have considerably improved the soundness of the manuscript in revision.  There are 2 issues that warrant mentioning. The first regards the choice of housekeeping/normalizing genes. The authors misunderstood the point I was trying to make. That is, both Actb and G3pdh are generally quite abundant both in vitro and in vivo. My concern was not about the stability of the housekeeping genes as indicated in the authors response to my review. Rather, because these mRNAs are abundant, they are more likely to skew the results of an RNA that is less abundant to magnify the difference,  One way to check to see if this is a problem, examine the Ct values of each test RNA with the RNA used for normalization (e.g. compare the Cts for Fkbp5, Nr3C1, Nr3c2, and FoxO1 with those corresponding to ActB in vivo and G3pdh, in vitro). If they are different by an order of magnitude, it would be worth considering a less abundant normalizing gene.

A second point which was not brought up in the initial review, the authors should consider listing a second limitation to the study. As the genes under investigation were chosen ahead of time, this study cannot be considered non-biased.  While there is nothing wrong with this approach, it should be stated up front.

Reviewer 2 Report

Comments and Suggestions for Authors

Accept in present form
